# Analysis of Clinical Parameters, Drug Consumption and Use of Health Resources in a Southern European Population with Diabetes That Did Not Contract COVID-19: A Longitudinal Big Data Study

**DOI:** 10.3390/ijerph19116835

**Published:** 2022-06-02

**Authors:** Ana Lear-Claveras, Bárbara Oliván-Blázquez, Ana Clavería, Sabela Couso-Viana, Rosa Magallón Botaya

**Affiliations:** 1Aragonese Research Group in Primary Care (Grupo Aragonés de Investigación en Atención Primaria/GAIAP), Aragon Health Research Institute, 50015 Zaragoza, Spain; analearc@gmail.com (A.L.-C.); rosamaga@unizar.es (R.M.B.); 2Department of Psychology and Sociology, University of Zaragoza, 50009 Zaragoza, Spain; 3Network for Research on Chronicity, Primary Care, and Health Promotion (RICAPPS), 08007 Barcelona, Spain; anaclaveriaf@gmail.com; 4I-Saúde Group, South Galicia Health Research Institute, 36201 Vigo, Spain; sabela.couso@iisgaliciasur.es; 5Vigo Health Area, SERGAS, 36201 Vigo, Spain; 6Department of Medicine, Psychiatry and Dermatology, University of Zaragoza, 50009 Zaragoza, Spain

**Keywords:** COVID-19, diabetes, lockdown, lifestyle, health resources

## Abstract

The lockdown measures imposed to stop the spread of the virus have affected the general population, but particularly people with chronic diseases such as diabetes. An observational real world data pre-post study of 86,615 individuals over the age of 16, having a medical history in the Aragon (Spain) Health Service and diagnosed with diabetes, without COVID-19 infection was undertaken. Clinical, pharmacological and health resource use variables were collected during the six months prior to the onset of the lockdown and during the six months after the lockdown ended. The Student’s *t*-test was used to analyse differences in means. Our study does not show clinically relevant changes six months following the end of the strict lockdown. The consumption, by these patients, of hypoglycaemic drugs and the use of health resources continue at below pre-pandemic levels, six months later. The interruption in care for these patients and the lifestyle change resulting from the pandemic do not appear to have had a significant impact on the health of the diabetic population.

## 1. Introduction

The current COVID-19 pandemic has tested the response and ability of people, governments and health systems around the world to adapt [1]. Its impact on essential health services is of great concern. According to the results of the PULSE survey carried out by the World Health Organisation (WHO) in 105 countries, the majority (90%) have suffered interruptions in essential healthcare services since the beginning of the pandemic [2]. These interruptions are likely to have serious adverse effects on the health of the most vulnerable populations, such as those with chronic diseases who need regular assistance and care, especially Primary Care (PC) [2,3].

These PC services, which helped people to manage chronic diseases before COVID-19, were modified with the arrival of the pandemic. Since then, they have been focused on detecting mild cases of the infection, following up positive cases and contact tracing. This interrupted their care of patients with chronic diseases to a greater or lesser extent. In addition, to reduce the risk of COVID-19 transmission, medical appointments were postponed and follow-up appointments were mostly carried out by telephone [4].

The restrictions imposed under the state of emergency to stop the spread of the virus has also had a direct impact on people with chronic diseases, and especially on those with diabetes. The global prevalence of diabetes was 9.3% in 2019 [5]. In Spain, the figure was 13.8%, counting type 2 diabetics alone [6]. The home lockdown measures applied have affected the diabetic population’s ability to lead a healthy lifestyle (balanced diet, physical activity, etc.), which plays a very important role in the proper management and control of this disease [7].

Previous studies about the impact that lifestyle modifications have had on patients with diabetes have had varying results. Some suggest that lockdown has not had a negative impact on the disease’s clinical parameters [8,9], while others report the deterioration of patients during the time when restrictions were imposed [10,11].

The deterioration of the clinical parameters of diabetes, and other non-communicable diseases such as cardiovascular diseases or cancer during the pandemic, could further increase the risk of poor prognosis and mortality from COVID-19 [12]. An association between the presence of diabetes and a more serious COVID-19 disease has been found by many studies [13,14].

Most studies on diabetes and COVID-19 have investigated the influence of this disease on the prognosis of infected patients, but fewer have analysed the impact of the pandemic on the uninfected diabetic population. These latest studies also mostly have cross-sectional designs and small sample sizes, and it is, therefore, necessary to carry out large-scale studies that provide a longitudinal perspective. 

Accordingly, this paper aims to study and conduct a longitudinal analysis of the possible changes in the clinical parameters, as well as in the hypoglycaemic drug consumption and the use of health resources, between the six months prior to the start of lockdown and the six months after the end of strict lockdown, in a southern European population with diabetes who did not contract COVID-19.

## 2. Materials and Methods

### 2.1. Design and Study Population

Observational real world data pre-post study of the population aged over 16 registered in the Autonomous Region of Aragon, in the north of Spain, with medical records in Primary Healthcare Centres (*n* = 1,122,151 people).

This study’s final sample is made up of individuals aged over 16 with a clinical history in the Aragon Health Service and a diagnosis of type 1 or type 2 diabetes, according to the International Classification of Primary Care (ICPC-2) criteria [15], who have not been infected with COVID-19 (*n* = 86,615). This last criterion was established to analyse changes in health status due exclusively to the measures imposed to control the pandemic. On 15 March 2020, the Spanish Government declared a state of national emergency, limiting mobility and requiring the population to stay at home until 3 May.

Data were collected from each individual at different time periods. The baseline measurement was taken in the six months prior to the start of lockdown (from 14 September 2019 to 15 March 2020), and the second measurement in the six months after the end of strict lockdown (until 4 November 2020).

### 2.2. Data Sources

This study is based on data from Aragon Primary Care centre’s longitudinal electronic health records.

Given that the healthcare system is universal, with practically no other Primary Care providers, it is considered that the data obtained in this study are representative of practically 100% of the population that met the study’s inclusion criteria.

### 2.3. Variables

The sociodemographic variables included in this study were: sex, age, pharmaceutical delivery and rurality of the health zones (defined as: rural—with less than 10,000 inhabitants—or urban—with more than 10,000 inhabitants). The number of deaths in the study population was also considered for each of the measurement periods.

Comorbidity with other chronic diseases was also considered. Those chronic conditions with a prevalence greater than 5% [16] were also collected (arrhythmias, heart failure, ischaemic heart disease, hypertension, dyslipidaemia, obesity, overweight, vein and artery disease, cerebrovascular disease, chronic bronchitis, chronic obstructive pulmonary disease (COPD), asthma, chronic kidney disease, hypo and hyperthyroidism, smoking, alcoholism, insomnia, anxiety and depression, autolytic attempt, anaemia, neoplasia, dementia, hearing loss, cataracts, glaucoma, osteoarthritis, osteoporosis and dorsopathy).

The variables of interest in our study are related to the clinical parameters of diabetes and its possible complications. The clinical parameters collected were: blood glucose level, glycated haemoglobin (HbA1c), blood creatinine and glomerular filtration. In addition, blood pressure measurements (systolic and diastolic), total cholesterol, low density lipoprotein (LDL), high density lipoprotein (HDL), triglycerides, weight and body mass index (BMI) were collected.

Patient drug use in the 6 months prior to the start of lockdown and in the 6 months after the end of strict lockdown was assessed through variations in the number of patients with diabetes who do not use hypoglycaemic drugs vis-à-vis those who consume one or more, and through the total number of defined daily doses (DDD) per 1000 inhabitants per day (DHD), dispensed by the pharmacy during each of the two periods. For the calculation of DHD, the Aragon population at the middle of each of the periods was used.
$$\mathrm{DHD}=\frac{\mathrm{Registered}\;\mathrm{consumption}\;\mathrm{of}\;\mathrm{the}\;\mathrm{active}\;\mathrm{ingredient}\;*\;1000\;\mathrm{inhabitants}}{\mathrm{Standard DDD}\;\times \;nº\mathrm{inhabitants}/\mathrm{period}\;*\;365\;\mathrm{days}}$$

It was decided to take the dispensed DHD and not the prescribed one, as some prescribed drugs may not be dispensed. According to the Anatomical Therapeutic Chemical classification system (ATC), the following pharmacological groups were assessed: A10A (insulins and analogues) and A10B (oral antidiabetics). 

Finally, these patients’ use of health system resources during the period under study was assessed using variables related to the use of PC services (number of nurse and general practitioner—GP—visits for ordinary or continuous care at a health centre or at home, and number of visits to other health centre professionals), and the use of specialised hospital services (number of visits to hospital’s specialised care, number of diagnostic tests performed, number of visits to accident and emergency—A&E—department, hospitalisations and admissions to intensive care unit—ICU—as well as the duration of these stays).

### 2.4. Statistical Analysis

Given our large sample size, we used parametric statistics [17]. To determine the characteristics of the population, and the total number of patients who take one or more hypoglycaemic drug, or none at all, a descriptive analysis of the study variables was carried out using frequencies (percentages) to summarise the categorical variables and measures of central tendency and dispersion (mean and standard deviation) for the continuous variables.

For the study population, mortality due to causes other than COVID-19 was assessed by calculating the crude mortality rate for each of the two periods. The Aragon population at the middle of each of the two periods was used as the denominator.

In order to ascertain the variations in clinical variables, if in any of the different periods of time (6 months prior to the start of lockdown or 6 months later) there was more than one measurement collected for the same clinical parameter, the median and the interquartile range (IQR) were calculated. For the two time periods, the mean and standard deviation (SD) of each of the clinical variables were calculated. To compare the difference of means between the two measurements a paired samples *t*-test was performed.

Differences in drug consumption were assessed through the DHDs dispensed in the pharmacy to the study population in each of the periods.

The mean and standard deviation of each variable of resource use (primary and hospital specialised care) were also calculated. A paired samples *t*-test was used to compare the difference of means too. For those variables with a fewer number of observations than 100, a Wilcoxon rank test was used.

The level of significance was established at 5% (*p* < 0.05). The statistical analysis was carried out using IBM SPSS Statistic 21 (IBM Corporation, New York, NY, USA) and R version 4.0.5 (IBM Corporation, New York, NY, USA) in a PC with 16 MB of RAM.

## 3. Results

On 14 September 2019, there were 86,615 people over 16 years of age, with a diagnosis of diabetes (type 1 or type 2) in their PC history, who, as of 3 November 2020, had not been infected with COVID-19. It should be noted that for the population aged over 16, the prevalence of diabetes was 7.72% at the beginning of 2019. The mean age of the sample was 69.5 years (SD: 13.7), with 48,436 men (55.9%) and 38,179 women (44.1%). Two-thirds (69.5%) had annual incomes below 18,000 euros, and more than half (51.2%) resided in urban areas (with more than 10,000 inhabitants) (Table 1). Hypertension (66.1%), followed by dyslipidaemia (58.2%) and dorsopathies (28.9%) were the most prevalent comorbid diseases among the population with diabetes under study. 

Of the individuals included in the study, 1887 died over the six months prior to the declaration of the state of emergency, and 2019 during the six months following the end of the lockdown. Taking into account the Aragon population during these two periods, the mortality rate per 1000 individuals was 1.4 [95%CI 1.4–1.5] and 1.5 [95%CI 1.5–1.6], respectively.

The variation in the clinical variables when comparing the two measurements, can be seen in Table 2. HbA1c parameters [*p* < 0.001, 95%CI: 0.04–0.11], total cholesterol [*p* < 0.001, 95%CI: 2.39–5.00], LDL cholesterol [*p* < 0.001, 95%CI: 2.61–4.84], weight [*p* < 0.001, 95%CI: 0.54–0.65] and BMI [*p* < 0.001, 95%CI: 0.17–0.24] show a significant trend towards a slight improvement. This is in contrast to the improvement observed in blood glucose and HDL cholesterol, which is only statistically significant for this last parameter [*p* 0.032, 95%CI: 0.04–0.94]. In the same table, we can observe a subtle deterioration for other variables such as blood creatinine [*p* < 0.001, 95%CI: −0.04–−0.02] and a decrease in the glomerular filtration rate [*p* < 0.001, 95%CI: 1.28–1.95]; both are statistically significant variations. For blood pressure six months after lockdown ended, only diastolic blood pressure experienced a statistically significant deterioration [*p* < 0.001, 95%CI: −0.42–−0.22]. 

In terms of drug use by patients with diabetes in the six months before the start of the lockdown and six months after lockdown, Table 3 shows an increase in the total number of patients who did not take any hypoglycaemic drugs [38,032 (43.9) vs. 40,568 (46.8)], and a decrease in the total number of patients who took one or more drugs. Table 4 also reveals a decrease in the total number of DHDs dispensed by pharmacies for all drugs included in the study, except for the insulins Aspart, Glargine and Degludec and the oral antidiabetic Metformin.

The use that the population with diabetes made of health resources can be seen in Table 5. Between the two measurement periods, the number of nurse and general practitioner visits decreased; with only the decrease in the number of nurse visits (continuous care), including those at the health centre and home visits, not being statistically significant. On the other hand, the same table shows an increase in the number of general practitioner visits (continuous care) at the health centre [*p* = 0.014, 95%CI: −0.14–−0.02], and the number of visits to other professionals, such as social workers [*p* = 0.020, 95%CI: −0.71–−0.06]. As for physiotherapy services, visits were considerably reduced compared to the baseline measurement [*p* = 0.006, 95%CI: 0.34–2.04].

In relation to hospital specialised care, we observe the opposite trend. While the number of first visits to specialised care increased in the six months after the end of strict lockdown, the number of follow-up consultations decreased; this was the only statistically significant decrease [*p* < 0.001, 95%CI: 0.07–0.12].

The performance of diagnostic tests also changed during the months under study. These variations were statistically significant for all tests [*p* < 0.05], except for resonances and retinographies.

Among these patients, the number of visits to A&E also decreased [*p* < 0.001, 95%CI: 0.06–0.16]. In terms of hospitalisations and ICU admissions, no statistically significant differences were observed in terms of the number of visits or the duration of these stays.

## 4. Discussion

Unlike the results published by other studies [10,11], our longitudinal, large-scale study does not show clinically relevant changes in the clinical parameters of diabetes, despite the observed interruptions in these patients’ care. These findings are consistent with the results of some previous studies that also report a neutral [18,19], or even a positive [7] influence of restrictive measures on glycaemic control in these patients. 

When patients with type 1 and 2 diabetes are analysed separately, the results are more heterogeneous. A systematic review that analysed 33 studies which included patients with type 1 (*n* = 33) and 2 (*n* = 8) diabetes showed significant improvements in glycaemic values, mainly in patients with type 1 diabetes [20]. In our study, we were unable to differentiate between patients with type 1 or type 2 diabetes, but we assumed that the vast majority of patients were type 2 diabetics because: (a) in Spain type 1 diabetes accounts for 1 in 10 cases of diabetes [21] and (b) the mean age of the sample was high. 

One of the factors that could explain the slight impact of lockdown in the clinical parameters shown in our study is the knowledge the patients with diabetes themselves have of the influence that their disease has on their prognosis if infected with COVID-19 [22]. This could have promoted better disease management among patients with diabetes (more frequent glycaemic control, awareness of taking medication, etc.) [23], and greater self-care [19]. This last would be reflected in the significant improvement in the total cholesterol, LDL, HDL, weight and BMI showed in our study. Some previous studies in this area have pointed to the possible improvement of dietary habits [24] and physical exercise [23,24] due to the lockdown in patients with diabetes.

This subtle improvement in clinical parameters could in turn explain the decrease in the consumption of hypoglycaemic drugs in the six months after the end of lockdown. Previous studies showed the same trend in terms of diabetes drug prescriptions [25,26].

The decrease in the consumption of these drugs could also be explained by the difficulty of accessing them. According to the results of the study carried out in our country by a Patient Organisation Platform (POP) [27], during the first wave of the pandemic, 79.3% of patients with chronic diseases found it difficult to access treatment. In the second wave, this percentage fell to 25.2%. 

During the first months of the pandemic, to avoid the spread of the new virus, people with diabetes in Aragon (as in other places [2,28]) had interruptions in primary health care. Nevertheless, they continued to have telephone consultations. This allowed these patients’ condition to be controlled and monitored [29], which could have prevented any short-term deterioration. Very similar results were reported by two studies [30,31] carried out in patients with type 2 diabetes. 

However, not all care for these patients can be done properly remotely. Patients with diabetes (especially elderly patients with type 2 diabetes) have a high prevalence of metabolic risk factors and comorbid conditions and, therefore, require regularly, face-to-face attention [32,33]. For these most fragile patients, telematic attention could have been more difficult during the months of the pandemic [30]. As shown in another work [34], the decrease in the number of nursing visits observed in our study could have represented a challenge in the nursing management of diabetic foot care during the COVID-19 pandemic. According to the results of a study [35] carried out in Catalonia (Spain), diabetic foot screening was the most affected indicator among the quality standards that decreased the most during the pandemic. 

Likewise, six months after the end of the strict home lockdown, the number of visits to specialised care and diagnostic tests conducted on patients with diabetes had not yet reached pre-pandemic levels. These results, are consistent with the results of other published works [2,28]. Diabetic retinopathy screening also showed a significant decrease in the study carried out in Catalonia [35]. In our study, the number of retinographies performed showed an opposite trend, however, these results were not statistically significant. According to the decrease in the number of visits to specialised care (successive consultations), another study conducted in England, showed a decrease in lower-limb major and minor amputation and revascularisation procedures among patients with diabetes in 2020 [36].

A mix between supply and demand was responsible for the disruption of services [2].

These interruptions could have caused a delay in the diagnosis and in the start of treatment, causing a worsening in clinical outcomes (decompensations or acute or chronic complications) in those more fragile individuals [2,28,33]. This fact could explain the increase in the number of deaths during the second measurement period, although this study does not include patients with diabetes who also contracted COVID-19, the excess of mortality due to other causes could have been caused by the concentration of resources in the fight against COVID-19. Another study also carried out in Spain reported similar results, showing a reduction in this trend in the first half of 2021 [37].

Our study has some limitations. Firstly, the dataset includes patients with both type 1 and type 2 diabetes, which means that it is dealing with patients with different clinical profiles. Secondly, for certain clinical variables, a limited number of records are available, because the GP must validate those results. Therefore, they are a sample of the real data. We cannot rule out the existence of a bias, although it is reasonable to consider that it would have the same size and direction in each of the periods considered. Thirdly, we do not know how long these patients have been diabetic. We also do not have information about the acute (hypoglycaemia, hyperglycaemia, diabetic ketoacidosis) and chronic (microvascular and macrovascular) complications of diabetes or the decompensation of other chronic diseases. Finally, this study also does not include self-reported data on disease management (glycaemic control, medication intake) and the lifestyle habits (diet, physical activity, work routine, etc.) maintained by the population under study during the months of lockdown. Conducting studies with a qualitative approach would provide information about the subjective perception that individuals have of the impact of lockdown on maintaining healthy lifestyles and managing their disease. Likewise, it would be interesting to evaluate the changes 12 months after the end of strict lockdown, to verify whether or not the clinical parameters follow the same trend over time. 

Episodes, use of health resources and drug consumption, are routinely collected from a variety of sources for the total population. Greater knowledge about the health demand and the real consequences of pandemics and their secondary effects, may improve health planning and resource management. A more detailed analysis of geographic variations will allow the identification of vulnerable populations. 

## 5. Conclusions

Our study contributes to the knowledge of the consequences of lockdown for the population of diabetes patients not infected with COVID-19 in a medium-size Spanish Health Authority. It offers a longitudinal perspective, considering variables related to clinical parameters, drug consumption and the use of health resources together. 

Our results suggest that diabetes patients without COVID-19 have been able to cope adequately with the restrictions imposed, with no clinically significant impact on their diabetes control.

## Figures and Tables

**Table 1 ijerph-19-06835-t001:** Sociodemographic data and chronic comorbidities in diabetic patients not infected with COVID-19.

	N (%)
Age Mean (SD)	69.5 (13.7)
Sex	
Men	48,436 (55.9)
Women	38,179 (44.1)
Pharmaceutical delivery	
<18,000	60,182 (69.5)
Between 18,000 and 100,000	22,140 (25.6)
>100,000	336 (0.4)
Free pharmacy	3189 (3.7)
Mutualist	706 (0.8)
Uninsured	62 (0.1)
Rurality of health zones	
Urban	44,321 (51.2)
Rural	42,293 (48.8)
Chronic comorbidities (Yes %)	
Arrhythmias	9266 (10.7)
Heart failure	5023 (5.8)
Ischaemic heart disease	9633 (11.1)
Hypertension	57,295 (66.1)
Dyslipidaemia	50,431(58.2)
Obesity	20,200 (23.3)
Overweight	1374 (1.6)
Vein/artery disease	4705 (5.4)
Cerebrovascular disease	7784 (9)
Chronic bronchitis	1354 (1.6)
COPD	5853 (6.8)
Asthma	4901 (5.7)
Chronic kidney disease	11,309 (13.1)
Hypothyroidism	9983 (11.5)
Hyperthyroidism	3932 (4.5)
Smoking	12,192 (14.1)
Alcoholism	1661 (1.9)
Insomnia	12,828 (14.8)
Anxiety and depression	23,289 (26.9)
Autolytic attempt	200 (0.2)
Anaemia	16,927 (19.5)
Neoplasia	23,682 (27.3)
Dementia	3725 (4.3)
Hearing loss	8914 (10.3)
Cataracts	15,300 (17.7)
Glaucoma	9838 (11.4)
Osteoarthritis	10,124 (11.7)
Osteoporosis	7772 (9.0)
Dorsopathy	25,051 (28.9)

(SD) Standard deviation; (COPD) Chronic Obstructive Pulmonary Disease.

**Table 2 ijerph-19-06835-t002:** Clinical parameters 6 months before and 6 months after lockdown.

		6 Months Before	6 Months After	Paired Samples *t*-Test	
	N	Mean (SD)		95%CI	*p*
Blood glucose level	2910	137.2 (38.6)	135.8 (41.2)	−0.06; 2.87	0.06
HbA1c (%)	2518	7.0 (1.1)	6.9 (1.1)	0.04; 0.11	<0.001
Blood creatinine	2688	0.9 (0.4)	1.0 (0.4)	−0.04; −0.02	<0.001
Glomerular filtration	2688	75.5 (22.0)	73.8 (22.5)	1.28; 1.95	<0.001
Systolic blood pressure	26,685	136.1 (14.3)	136.2 (15.4)	−0.33; 0.01	0.073
Diastolic blood pressure	26,691	75.6 (8.8)	75.9 (9.3)	−0.42; −0.22	<0.001
Total cholesterol	2708	179.6 (39.3)	175.9 (38.9)	2.39; 5.00	<0.001
LDL	2459	100.2 (33.3)	96.5 (31.6)	2.61; 4.84	<0.001
HDL	2638	50.8 (16.0)	50.3 (13.2)	0.04; 0.94	0.032
Triglycerides	2669	145.9 (84.0)	147.5 (84.5)	−4.08; 0.94	0.211
Weight	17,095	78.3 (15.0)	77.7 (15.2)	0.54; 0.65	<0.001
BMI	10,800	29.9 (5.0)	29.7 (5.1)	0.17; 0.24	<0.001

(HbA1c) Glycated haemoglobin; (LDL) Low density lipoprotein; (HDL) High density lipoprotein; (BMI) Body mass index; (SD) Standard deviation; (95%CI) Confidence interval.

**Table 3 ijerph-19-06835-t003:** Number of patients who do not use insulins or oral antidiabetics or who consume one or more.

	6 Months Before	6 Months After
	N (%)	N (%)
No hypoglycaemic drug	38,032 (43.9)	40,568 (46.8)
One hypoglycaemic drug	36,360 (42.0)	34,461 (39.8)
Two hypoglycaemic drugs	10,325 (12.0)	9968 (11.5)
Three or more hypoglycaemic drugs	1898 (2.1)	1618 (1.9)

**Table 4 ijerph-19-06835-t004:** Number of DHDs 6 months before and 6 months after lockdown.

		6 Months Before	6 Months After
	N	DHD	
Insulins			
Insulin (human) fast acting	655	0.63	0.61
Lispro fast acting	2106	4.15	4.15
Aspart fast acting	5490	9.13	9.34
Glulisine	1179	2.05	1.76
Insulin (human) intermediate acting	448	1.01	0.98
Insulin (human) intermediate acting + fast acting	255	0.70	0.65
Lispro intermediate acting	922	2.71	2.61
Aspart intermediate acting	1591	5.10	4.73
Glargine	14,491	22.70	23.10
Determir	1620	3.26	3.18
Degludec	3060	4.44	4.74
Oral antidiabetics			
Metformin	31,076	39.96	42.30
Glibenclamide	289	0.46	0.42
Glipizide	110	0.21	0.18
Gliclazide	2369	4.25	3.90
Glimepiride	1951	7.26	6.57
Glisentide	5	0.01	0.01

(DHDs) Defined Daily Doses per 1000 inhabitants per day.

**Table 5 ijerph-19-06835-t005:** Number of visits and diagnostic tests prescribed 6 months before and 6 months after lockdown.

		6 Months Before	6 Months After	Paired Samples *t*-Test	
	N	Mean (SD)		95%CI	*p*
No. of nursing visits (ordinary care) at health centre or by telephone	54,995	5.26 (5.27)	4.53 (4.85)	0.68; 0.76	<0.001
No. of nursing visits (ordinary care) at home	4985	7.03 (9.94)	6.70 (9.80)	0.04; 0.60	0.023
No. of nursing visits (continuous care) at health centre	2108	2.31 (3.53)	2.19 (3.57)	−0.03; 0.27	0.112
No. of nursing visits (continuous care) at home	621	2.30 (2.99)	2.28 (3.67)	−0.30; 0.35	0.861
No. of general practitioner visits (ordinary care) at health centre or by telephone	68,803	5.36 (4.24)	5.29 (4.72)	0.03; 0.10	<0.001
No. of general practitioner visits (ordinary care) at home	2811	3.44 (3.67)	2.80 (3.33)	0.51; 0.77	<0.001
No. of general practitioner visits (continuous care) at health centre	4458	1.85 (1.82)	1.93 (2.19)	−0.14; −0.02	0.014
No. of general practitioner visits (continuous care) at home	682	1.84 (1.63)	1.68 (1.38)	0.03; 0.28	0.014
No. of visits to other professionals					
Physiotherapist	294	5.63 (6.13)	4.43 (6.16)	0.34; 2.04	0.006
Midwife	141	2.45 (2.16)	2.52 (2.06)	−0.53; 0.37	0.732
Dentist	257	2.15 (1.82)	2.41 (1.88)	−0.55; 0.04	0.092
Social worker	426	2.75 (2.65)	3.13 (3.45)	−0.71; −0.06	0.020
No. of visits to specialised care (first consultation)	3105	1.49 (0.87)	1.53 (0.93)	−0.08; 0.00	0.067
No. of visits to specialised care (successive consultations)	26,307	2.71 (2.27)	2.61 (2.38)	0.07; 0.12	<0.001
No. of diagnostic tests performed					
X-rays	12,895	1.18 (1.34)	1.03 (1.26)	0.12; 0.18	<0.001
Ultrasound	12,895	0.33 (0.58)	0.30 (0.54)	0.01; 0.04	<0.001
Resonance	12,895	0.11 (0.36)	0.12 (0.35)	−0.01; 0.00	0.148
CT scans	12,895	0.36 (0.67)	0.37 (0.68)	−0.03; −0.00	0.028
Retinographies	41	1.02 (0.16)	1.15 (0.65)	−0.33; 0.09	0.257 ^a^
Other imaging test	12,895	0.15 (0.43)	0.14 (0.41)	0.00; 0.02	0.019
Haemograms	22,398	0.31 (0.54)	0.22 (0.48)	0.08; 0.09	<0.001
Biochemistry	22,398	1.05 (0.65)	0.98 (0.64)	0.05; 0.07	<0.001
Microbiology	22,398	0.18 (0.56)	0.24 (0.64)	−0.07; −0.05	<0.001
Immunology test	22,398	0.14 (0.40)	0.13 (0.38)	0.01; 0.02	<0.001
Coagulation	22,398	0.03 (0.17)	0.04 (0.20)	−0.01; −0.00	<0.001
Urine test	22,398	0.40 (0.61)	0.36 (0.60)	0.03; 0.05	<0.001
No. of visits to the emergency department	4333	1.78 (1.50)	1.66 (1.29)	0.06; 0.16	<0.001
No. of hospital admissions	1335	1.54 (1.04)	1.54 (1.05)	−0.07; 0.06	0.926
No. of days of hospital stay	1335	18.70 (56.44)	17.37 (45.54)	−0.30; 2.95	0.111
No. of ICU admissions	5	1 (0)	1 (0)		(*) ^a^
No. of days of ICU stay	5	22.20 (27.36)	18.60 (27.59)	−5.20; 12.40	0.465 ^a^

(*) The correlation and t cannot be calculated because the standard error of the differences is 0. ^a^ Wilcoxon signed-rank test. (ICU) Intensive care unit; (SD) Standard deviation; (95%CI) Confidence interval.

## Data Availability

This report does not contain patient identifiable data. Consent from individuals involved in this study was not required. Requests for any underlying data cannot be granted by the authors because the data was acquired under a license/data sharing agreement with the Aragon Health Services, under which conditions of use (and further use) apply.

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
