# Peer review of "Analysis of Clinical Parameters, Drug Consumption and Use of Health Resources in a Southern European Population with Diabetes That Did Not Contract COVID-19: A Longitudinal Big Data Study"

_ijerph, 2022, doi:10.3390/ijerph19116835_

Round 1

Reviewer 1 Report

Firstly, I thank you authors for their study. It contains some important information and the results are interesting. However, the authors did not discuss their results properly in their discussion section.

The discussion section must be completely rewritten as it is not a discussion currently. The first 5 paragraphs of the section are like an introduction and then they mentioned their results in the 6th paragraph. Afterward, they mentioned the limitations in the 8th paragraph (which are incomplete as well).

The conclusion should be more focused and also indicate any potential limitations and future perspectives without speculations. 

Author Response

COMMENT 1. Firstly, I thank you authors for their study. It contains some important information and the results are interesting. However, the authors did not discuss their results properly in their discussion section.

The discussion section must be completely rewritten as it is not a discussion currently. The first 5 paragraphs of the section are like an introduction and then they mentioned their results in the 6th paragraph. Afterward, they mentioned the limitations in the 8th paragraph (which are incomplete as well).

As you suggest, we have rewritten this section.

COMMENT 2. The conclusion should be more focused and also indicate any potential limitations and future perspectives without speculations. 

Following your recommendations, we have also rewritten this section.

Reviewer 2 Report

This article is another one of the same project, similar to other published articles. It is interesting and contemporary, but needs some corrections.

Some suggestions for correction.

1. There are 6 groups of pharmaceutical delivery in the Table 1. Probably not all groups belong to pharmaceutical delivery.

2. Urban and semi-urban/rural are not health areas.

3. Two right-side columns in the Table 2 and Table 5 must have name.

4. Statistical comparisons of numbers before and after lockdown in the Table 3 and Table 4 are advisable.

5. Distribution of some variables presented in the Table 5 is not normal or even far from normal. Therefore the parametric test is not appropriate for comparison, nonparametric test should be applied.

6. Doubtful use of the term “big data” in the title, because evaluation of changes in most clinical parameters and drug consumption parameters of less than 10% patients was made. More discussion about significance of the results obtained from small part of patients could be written.

Author Response

COMMENT 1. There are 6 groups of pharmaceutical delivery in the Table 1. Probably not all groups belong to pharmaceutical delivery.

This variable was considered a proxy for income and, therefore for socioeconomic status.

The copayment in health was created with the aim of making the user aware of the cost of medicines and avoiding inappropriate consumption.

 In Spain, with the entry into force of Royal Decree Law 16/2012, of April 20, the pharmaceutical copay changed to modulate based on criteria of income, employment status and chronicity (Table 1). Since its launch, the unfair situations it causes have been denounced, especially among retirees, who are the most affected group.

Code

Type of contribution

Insured and beneficiaries

TSI 001

Free pharmacy

a) People with disabilities.

b) Persons receiving social integration income.

c) Persons receiving non-contributory pensions.

d) Unemployed who do not receive an unemployment benefit.

e) Treatments derived from work accidents and professional illness.

TSI 002 - 01

10%

Pensioners with incomes below 18,000€ per year.

TSI 002 - 02

10%

Pensioners with incomes between 18,000 and 100,000€ per year.

TSI 003

40%

Working age population with incomes below 18,000€ per year.

TSI 004

50%

Working age population with incomes between 18,000 and 100,000€ per year.

TSI 005

60%

Working age population with incomes above 100,000€ per year.

TSI 005 - 03

60%

Pensioners with incomes above 100,000€ per year.

TSI 006

30%

Mutualist.

TSI* Individual Health Card.

COMMENT 2. Urban and semi-urban/rural are not health areas.

Thank you for your comment.

The basic health zone is the one with the most basic geographical delimitation used as a reference to plan and organize the work of Primary Care teams. These teams are made up of health and non-health professionals who are in charge of caring for the population of the basic health zone. For rurality, according to demographic criteria, the official statistics establish the cut-off point to differentiate urban or rural municipalities in 10,000 inhabitants.  

We have changed it to clarify this aspect as follow:

  • 3 Variables: “The sociodemographic variables included in this study were: sex, age, pharmaceutical delivery and rurality of the health zones (defined as: rural – with less than 10,000 inhabitants – or urban – with more than 10,000 inhabitants –.”
  • Table 1: “Rurality of health zones.”

COMMENT 3. Two right-side columns in the Table 2 and Table 5 must have name.

As you suggest, we have added “Paired samples t-test” in Table 2 and in Table 5.

COMMENT 4. Statistical comparisons of numbers before and after lockdown in the Table 3 and Table 4 are advisable.

We haven´t reported the p-values in Table 3 and Table 4 because we calculated the difference in drug consumption through the real DHDs dispensed in the pharmacy to the study population (which corresponds to the entire diabetic population in Aragon) in the two periods.

COMMENT 5. Distribution of some variables presented in the Table 5 is not normal or even far from normal. Therefore the parametric test is not appropriate for comparison, nonparametric test should be applied.

Due to the large sample size, parametric test were deemed appropriate, since in large samples even if the data distribution is not normal, statistics tend to be normal [1].

As we have explained in the statistical analysis section for those variables with a fewer number of observations than 100, in Table 5 a Wilcoxon rank test was used. We appreciate the reviewer’s observation and we have added this clarification as a footnote in Table 5.

[1] Lubin Pigouche P, Maciá Antón MA, Rubio de Lemus P. Mathematical Psychology. Madrid: Universidad Nacional de Educación a distancia; 2005.

COMMENT 6. Doubtful use of the term “big data” in the title, because evaluation of changes in most clinical parameters and drug consumption parameters of less than 10% patients was made. More discussion about significance of the results obtained from small part of patients could be written.

Thank you for pointing it out. We have the term “big data” in the title because a combination of different databases were performed, including: population individual identification (3 million records), primary care electronic health record, hospital discharge, diagnostic tests, imaging and prescription. Although, as the reviewer points out, there was an evaluation of less than 10% of patients, it is also true that the other 90% were also analyzed. In fact, 1,122,151 records with almost 300 variables were analyzed. Through a detailed literature review, Baro et al [2] provide a current and quantitative definition of big data: a dataset with Log(n*p) superior or equal to 6. In our study, that threshold is clearly overcome.

[2] Baro, E., Degoul, S., Beuscart, R., & Chazard, E. (2015). Toward a literature-driven definition of big data in healthcare. BioMed research international, 2015.

As suggested, we have completed the discussion as follow:

“Secondly, for certain clinical variables, a limited number of records are available, because GP must validate those results. Therefore, they are a sample of the real data. We cannot rule out the existence of a bias, although it is reasonable to consider that it would have the same size and direction in each of the periods considered.”

“Episodes, use of health resources and drug consumption, are routinely collected from a variety of sources for the total population. Greater knowledge about the health demand and the real consequences of pandemics and their secondary effects, may improve health planning and resource management. A more detailed analysis of geographic variations will allow the identification of vulnerable populations.”

Reviewer 3 Report

The Manuscript entitled: Analysis of clinical parameters, drug consumption, and use of health resources in a Southern European population with diabetes that did not contract COVID-19: a longitudinal big data study has been reviewed.

Minor Queries/Comments:

  • Discussion section: Authors should focus more on the main findings and avoid repeating the results presented in the discussion. Clinical relevance should be added. Authors could also correlate their findings with what has been published in the literature.
  • Statistical analysis: Please indicate the software's name employed, the level of significance, and the name of the normality test used.
  • Could you please indicate where (which table(s)) the data were presented as median (IQR)? Please revise.
  • Please report the p-values in Table 4.
  • Please clarify for each p-value reported whether you had used a paired comparison test or a Wilcoxon signed-rank test.

Author Response

COMMENT 1. Discussion section: Authors should focus more on the main findings and avoid repeating the results presented in the discussion. Clinical relevance should be added. Authors could also correlate their findings with what has been published in the literature.

According to your suggestions, we have rewritten this section.

COMMENT 2. Statistical analysis: Please indicate the software's name employed, the level of significance, and the name of the normality test used.

We appreciate the reviewer's observation and we have added the next sentences in the statistical analysis section:

The level of significance was established at 5% (p <0.05). The statistical analysis was carried out using IBM SPSS Statistic 21 and R version 4.0.5 in a PC with 16 MB of RAM.”

We didn´t report normality test used because we didn´t use it. Due to the large sample size, parametric test were deemed appropriate, since in large samples even if the data distribution is not normal, statistics tend to be normal [1].

[1] Lubin Pigouche P, Maciá Antón MA, Rubio de Lemus P. Mathematical Psychology. Madrid: Universidad Nacional de Educación a distancia; 2005.

COMMENT 3. Could you please indicate where (which table(s)) the data were presented as median (IQR)? Please revise.  

For each individual in all clinical variables we took one measurement in each period. In all variables included in Table 2, some individuals had more than one measurement in each period, so in those cases we calculated the median. When all individuals had only one measurement we calculated the mean as we showed in Table 2.

COMMENT 4. Please report the p-values in Table 4.  

We haven´t reported the p-values in Table 4 because we calculated the difference in drug consumption through the real DHDs dispensed in the pharmacy to the study population (which corresponds to the entire diabetic population in Aragon) in the two periods.

COMMENT 5. Please clarify for each p-value reported whether you had used a paired comparison test or a Wilcoxon signed-rank test.

We appreciate the reviewer's observation. We have added it as footnote in Table 5 in those variables with less number of observations than 100:

a Wilcoxon signed – rank test.

Round 2

Reviewer 1 Report

I am happy with the improvement in the discussion (and conclusion) section. Just two minor comments: 

In line 370: "...." should be removed.

All sentences/phrases should be complete. Overall the formatting should be uniform and within the journal's requirements. Please do a formatting check. Thank you.

Author Response

Thank you very much for your comments.

Changes has been done.